# The Protective Role of Microglial PPARα in Diabetic Retinal Neurodegeneration and Neurovascular Dysfunction

**DOI:** 10.3390/cells11233869

**Published:** 2022-12-01

**Authors:** Tian Yuan, Lijie Dong, Elizabeth A. Pearsall, Kelu Zhou, Rui Cheng, Jian-Xing Ma

**Affiliations:** 1Department of Biochemistry, Wake Forest University School of Medicine, Winston-Salem, NC 27157, USA; 2Tianjin Key Laboratory of Retinal Functions and Diseases, Tianjin Branch of National Clinical Research Center for Ocular Disease, Eye Institute and School of Optometry, Tianjin Medical University Eye Hospital, Tianjin 300384, China; 3Vision Research Center, University of Missouri-Kansas City, Kansas City, MO 64108, USA

**Keywords:** diabetic retinopathy, microglial metabolism, mitochondrion, glycolysis, inflammation, neurodegeneration, pericyte loss

## Abstract

Microglial activation and subsequent pathological neuroinflammation contribute to diabetic retinopathy (DR). However, the underlying mechanisms of microgliosis, and means to effectively suppress pathological microgliosis, remain incompletely understood. Peroxisome proliferator-activated receptor alpha (PPARα) is a transcription factor that regulates lipid metabolism. The present study aimed to determine if PPARα affects pathological microgliosis in DR. In global *Pparα* mice, retinal microglia exhibited decreased structural complexity and enlarged cell bodies, suggesting microglial activation. Microglia-specific conditional *Pparα*^−/−^ (PCKO) mice showed decreased retinal thickness as revealed by optical coherence tomography. Under streptozotocin (STZ)-induced diabetes, diabetic PCKO mice exhibited decreased electroretinography response, while diabetes-induced retinal dysfunction was alleviated in diabetic microglia-specific *Pparα*-transgenic (PCTG) mice. Additionally, diabetes-induced retinal pericyte loss was exacerbated in diabetic PCKO mice and alleviated in diabetic PCTG mice. In cultured microglial cells with the diabetic stressor 4-HNE, metabolic flux analysis demonstrated that *Pparα* ablation caused a metabolic shift from oxidative phosphorylation to glycolysis. *Pparα* deficiency also increased microglial STING and TNF-α expression. Taken together, these findings revealed a critical role for PPARα in pathological microgliosis, neurodegeneration, and vascular damage in DR, providing insight into the underlying molecular mechanisms of microgliosis in this context and suggesting microglial PPARα as a potential therapeutic target.

## 1. Introduction

Diabetic retinopathy (DR) remains a leading cause of blindness in the United States [1]. Although DR is considered a neurovascular disease, increasing evidence has identified chronic inflammation, including pathological activation of innate immunity, as an important contributor to DR pathogenesis [2]. Clinical studies have demonstrated that serum inflammatory markers are increased in diabetic patients [3]. Importantly, inflammatory cytokine and chemokine levels are significantly higher in vitreous samples from DR patients relative to non-diabetic control subjects [4]. It is thus essential to investigate the role of inflammation in DR and the underlying molecular mechanisms.

In the retina, microglia function as both glial cells and the major resident innate immune cell, so they are critical to immunity and maintenance of retinal homeostasis. During inflammation, microglia release inflammatory factors, including cytokines and chemokines, that interact with multiple cell types and initiate a cytotoxic or cytoprotective response, depending on the microglial subpopulation and activation state [5]. Interestingly, retinal microglia activation occurs in patients with proliferative DR and diabetic macular edema [6,7]. Inhibition of microglial activity decreases retinal neural apoptosis in diabetic rats [5]. Together, these observations suggest a pivotal role of microglia-mediated inflammation in the pathogenesis of diabetic microvascular complications.

Microglial activation is strongly associated with metabolic reprogramming, such as a shift from mitochondrial oxidative phosphorylation to glycolysis. Recent studies have demonstrated critical connections between microglial activation and reprogramming of microglial metabolic profiles in the brain [8,9,10]. Activated microglia undergo polarization towards both inflammatory (M1-like) and anti-inflammatory (M2-like) phenotypes and have distinct functional roles depending on their activation state [2]. Generally, M1-like microglia release pro-inflammatory cytokines such as TNFα to induce inflammation and neurotoxicity, while M2-like microglia secrete anti-inflammatory cytokines such as IL-10, TGF-β, and IGF-1 to inhibit inflammation and confer neuroprotection [11,12]. Microglia utilize both glycolysis and mitochondrial oxidative energy metabolism for ATP production. Recent studies have demonstrated that quiescent and M2-like microglia primarily use oxidative phosphorylation to generate ATP, whereas M1-like microglia shift to glycolytic metabolism, allowing more rapid ATP production with no oxidative investment [12]. Accordingly, disruption of metabolic homeostasis alters microglial function in pathological conditions, although the underlying molecular mechanisms for disrupting microglial metabolic homeostasis remain incompletely understood [9,13,14]. Diabetes is a metabolic disease characterized by deficient systemic glucose metabolism and subsequent localized metabolic disruptions in tissue microenvironments [15]. In the present study, we sought to investigate metabolism-associated retinal microglial functional changes and the contributions of these pathologies to retinal vascular damage and neurodegeneration in DR.

Peroxisome proliferator-activated receptor alpha (PPARα) is a key regulator of lipid oxidation. Clinical studies reported unexpectedly that the PPARα agonist fenofibrate, which is traditionally used to treat hypertriglyceridemia and primary hypercholesterolemia, has robust therapeutic effects on DR in type 2 diabetic patients, and the underlying mechanisms of action remain incompletely understood [16,17,18,19,20]. In the brain, PPARα agonists inhibit microglial inflammation and decrease the release of pro-inflammatory cytokines by suppressing NF-κB activity [21,22,23]. Previous studies from our laboratory have reported beneficial roles of PPARα in retinopathy and provided some insight into its mechanisms of action [24,25,26,27]. However, the potential involvement of PPARα in the regulatory cascade of retinal microglia polarization and its potential effects on DR-associated inflammation in this context remains unknown. To address these research questions, we generated microglia-specific conditional *Pparα* knockout mice (PCKO) and transgenic mice with microglia-specific *Pparα* overexpression (PCTG). Together with assessing microglial metabolism under diabetic stress in the presence and absence of *Pparα* expression, we have provided insight into the role of microglial PPARα in retinopathy under diabetic conditions. 

## 2. Materials and Methods

### 2.1. Animals and Diabetes Induction

Global *Pparα* knockout mice (*Pparα*^−/−^) were purchased from The Jackson Laboratory (Jackson Laboratory, Bar Harbor, ME, USA). *Pparα^flox/flox^* mice in which *Pparα* exon 4 was flanked with loxP sites, were generated through a contracted service with Ingenious Targeting Laboratory (Ingenious Targeting Laboratory, Ronkonkoma, NY, USA). *Pparα* transgenic mice that express *Pparα* under the chicken β-actin promoter upon removal of a floxed stop cassette by the desired tissue-specific Cre recombinase were generated through a contract service with Cyagen Biosciences (Cyagen Biosciences, Santa Clara, CA, USA). These mice were crossbred with *Cx3Cr1^CreERT2^* (Jackson Laboratory, Bar Harbor, ME, USA) mice to generate microglia-specific conditional *Pparα* knockout mice (PCKO) and microglia-specific *Pparα* transgenic (PCTG) mice. All animals were bred into the C57BL/6J background under pathogen-free conditions and tested negative for the Rd8 (Crb1) mutation. At experimental endpoints, mice were euthanized by carbon dioxide asphyxiation. All experiments were approved by the Institutional Animal Care and Use Committee of Wake Forest University.

### 2.2. Tamoxifen Induction

Cre recombinase activity in PCKO and PCTG mice was induced by daily intraperitoneal (i.p.) injections of 75 mg tamoxifen/kg body weight dissolved in corn oil for 5 consecutive days from 3 weeks of age [28]. 

### 2.3. Streptozotocin (STZ)-Induced Diabetes

Eight to ten-week-old mice received 55 mg/kg STZ via i.p. injections for 5 consecutive days. Blood glucose levels were measured 7 days following the final injection and monthly thereafter. Only animals with hyperglycemia (blood glucose > 350 mg/dL) were included as diabetic mice in experiments. 

### 2.4. Pericytes Quantification

Retinal pericytes were quantified in mice at the age of 8–9 months. As previously described, the pericytes were counted in flat-mounted retinas with periodic acid-Schiff (PAS) staining [29]. In brief, the retinas were dissected. Followed 4% PFA fixation, the retinas were incubated in 3% trypsin digestion solution at 37 °C. The vasculature was separated with water dropping and stained with PAS (Sigma-Aldrich, St. Louis, MO, USA). The retinal vasculature was imaged under an Olympus BX43 microscope (Olympus, Tokyo, Japan). 

### 2.5. Electroretinogram Recording

Retinal function was evaluated by electroretinogram (ERG) recording on mice about 8–9 months of age [30]. Briefly, animals were dark-adapted for at least 12 h before the scotopic (Rod) and photopic (Cone) ERG assessment. Animals were anesthetized by injecting a mixture of ketamine/Xylazine after pupil dilation with 1% cyclopentolate hydro-chloride ophthalmic solution (Alcon Laboratories, Inc, Fort Worth, TX, USA). Gonak (Akorn, Lake Forest, IL, USA) was used as a corneal lubricant. Animals were placed on a heated ERG platform to maintain body temperature under anesthesia. Electrodes were placed as follows: a pair of gold wire loop electrodes was placed on the eye, a reference hook electrode was placed on the cheek, and a ground needle electrode was placed on the tail. The ERG response was recorded with a Diagnosys Espion Visual Electrophysiology System (Diagnosys LLC, Lowell, MA, USA) using light flashes with an intensity of 200 cd.s/m^2^ for scotopic ERG amplitude and 600 cd.s/m^2^ for photopic ERG amplitude. 

### 2.6. Fundoscopy and Optical Coherence Tomography (OCT) Imaging

Mouse pupils were dilated under anesthesia, as described above. Fundus images were obtained at 8–9 months of age using a Micron IV retina-imaging microscope (Phoenix Research Labs, Pleasanton, CA, USA). As previously described, OCT images were captured using an Envisu R2000 system (Bioptigen, Durham, NC, USA) [30]. Total retinal thickness was measured as the distance from the inner retinal nerve fiber layer to the outer retinal pigment epithelial layer using Bioptigen Diver (Version 3.4.4.0, Bioptigen, Morrisville, NC, USA). 

### 2.7. Immunofluorescence and Analysis

Retinas were dissected and blocked in PBS with 10% goat serum and 0.3% Triton X-100 for 2 h following 4% PFA fixation for 1 h. Flat-mounted retinas were incubated with primary antibody against anti-Iba1 in blocking solution overnight at 4 °C, followed by incubating with a secondary antibody for 3 h (Appendix A). Images were captured using an Olympus FV1200 SPECTRAL laser scanning confocal microscope (Olympus, Shinjuku City, Tokyo, Japan). For morphology analysis, Z-stack images were taken (10 µm thickness) under the 40× objective. Images were skeletonized, and cell endpoints and total branch length were quantified using the Analyze Skeleton plugin for Fiji ImageJ software (version v1.53p), as previously described [31]. To quantify microglia soma size, cell processes were removed after image thresholding. Soma size was measured using the *Analyze particles* function in Fiji ImageJ. All the microglia on each image were analyzed for morphology to avoid bias. A total of 11–29 microglia were analyzed for each retina from an individual mouse.

After fixation for 1 h in 4% PFA, eyecups with the retina removed were dissected for subretinal flat-mount staining. RPE/choroid flat mounts were blocked with 5% BSA and 0.3% Triton X-100 in PBS for 1 h. The immunostaining was performed. Subretinal microglia numbers on each RPE/Choroid flat mount were counted under a Zeiss AxioObserver Z1 epi-fluorescent microscope (Carl Zeiss, Göttingen, Germany).

Mouse eyeballs were fixed in Davidson’s Fixative to prepare 6-µm paraffin-embedded sections or in 4% PFA for 8-µm cryosections. Sections were incubated with primary antibodies overnight at 4 °C (Appendix A), followed by the secondary antibody incubations with DAPI (1:10,000, Sigma) for 2 h. For microglial density, all cells in which the entire soma were presented were counted. Microglial STING expression was quantified and calculated as the percentage of STING and CD11b co-localized pixels/CD11b pixels using the *Colocalization threshold* plugin in Fiji ImageJ. Five to twelve microglia were analyzed for each animal. HardSet™ Antifade Mounting Medium (Vector Laboratories, Inc., Newark, CA, USA) was used for flat mounts and sections to preserve the fluorescence signal. 

### 2.8. Cell Models

The Human microglia HMC3 cell line was purchased from ATCC (CRL-3304, Manassas, VA, USA) and cultured in MEM with 10% FBS and 1% antibiotic-antimycotic. 

Primary microglial cells were prepared following a previously published method with minor modifications [32,33]. In brief, L929 cells from ATCC were cultured for 10 days without changing the medium. Conditioned medium was collected, sterilized, filtered, and stored at −80 °C for primary microglia culture. Brain tissue was harvested from neonatal mice (Postnatal Day 1–3) and digested with collagenase D (200 µg/mL, Sigma Aldrich, St. Louis, MO, USA) in FBS-free DMEM. Single cells were obtained by passing the cell suspension through a 40-µm cell strainer. Following centrifugation, mixed cells were resuspended in microglia primary culture medium (DMEM/F12 with 20% FBS, 20% L929 conditioned medium, and 1% antibiotic-antimycotic and 1% glutamine) for 2 days. On day 3, medium was replaced with fresh primary culture medium containing 50 ng/mL GM-CSF (315-03, PeproTech, Cranbury, NJ, USA). On day 10, microglial cells were isolated from mixed glial cultures via mechanical shaking at 180 rpm for 2 h. 

### 2.9. RNA Interference 

ON-TARGETplus human *Pparα* siRNA SMARTPool and scrambled control siRNAs (Dharmacon Inc., Lafayette, CO, USA) were transfected into HMC3 cells with HiPerFect transfection reagent (QIAGEN, Germantown, MD, USA) following the manufacturer’s instruction. 

### 2.10. OXPHOS and Glycolysis Analysis

The mitochondrial (Mito) stress assay and the Glycolytic Rate assay were conducted using a Seahorse XF Pro Extracellular Flux Analyzer (Agilent Technologies, Santa Clara, CA, USA) following manufacturer protocols. For the Mito stress assay, oxygen consumption rate (OCR) was recorded after injection of oligomycin (1.5 µM), FCCP (1.5 µM), and rotenone/antimycin A (RAA, 0.5 µM) sequentially. Glycolytic proton efflux rate (glycoPER) was calculated after injections of rotenone/antimycin A (0.5 µM) and 2-deoxyglucose (50 mM). 

### 2.11. Western Blot Analysis

Cells were lysed with SDS loading buffer containing protease and phosphatase inhibitors (Pierce–Thermo Scientific, Rockford, IL, USA). Protein concentration was determined using a BCA Protein Assay kit (Thermo Scientific, Waltham, MA, USA). Cellular proteins were separated by SDS-PAGE and transferred onto a 0.2 µM PVDF membrane, which was blocked for 1 h with 5% (*w*:*v*) milk. After incubation overnight with a primary antibody at 4 °C, the membrane was washed 3 × 15 min in TBS-T and incubated with HRP-conjugated secondary antibody for 2 h (Appendix A). Chemiluminescent signals were developed by incubation with Pierce ECL Western Blotting Substrate (Thermo Scientific, Waltham, MA, USA). Images were acquired using Chemidoc (Bio-Rad, Hercules, CA, USA), and the band density was measured with Fiji ImageJ. 

### 2.12. Data Analysis

Statistical analyses were performed and generated with GraphPad Prism 9.0 (GraphPad Software, San Diego, CA, USA). Comparison between two groups was analyzed using an unpaired two-tailed Student’s *t*-test. Comparison among multiple groups was analyzed via one-way ANOVA followed by Tukey's post hoc analysis. A *p*-value < 0.05 was considered statistically significant. All data are presented as the mean ± SD.

## 3. Results

### 3.1. Increased Microglial Density and Activation in the Ppara^−/−^ Retinas

To examine the role of PPARα in regulating retinal microglial activation, retinal microglia from *Ppara^−/−^* mice were stained with the microglial marker Iba1. Microglial cell density was quantified, revealing that at 9 months of age, microglial density was significantly increased in *Ppara^−/−^* retinas relative to wild-type (WT) controls (Figure 1A,B). In addition to the increased cell number in gliosis, microglial activation is often accompanied by morphological changes. Activated microglia have shortened process lengths, decreased complexity, and enlarged, ameboid cell bodies [31,34]. Therefore, we quantified microglial morphological complexity via skeleton analysis (Figure 1C–F). Microglia in *Pparα*^−/−^ retinas exhibited decreased structural complexity accompanied by decreased total branch length (Figure 1G) and endpoints (Figure 1H). In addition, *Pparα*^−/−^ microglia exhibited enlarged soma size relative to WT controls (Figure 1I–K). Taken together, these findings demonstrated microglial activation and polarization in *Pparα*^−/−^ retinas, suggesting that ablation of endogenous *Pparα* caused retinal microgliosis. 

To specifically study the role of PPARα in microglia and exclude the effects of surrounding cells, such as photoreceptors, vascular endothelial cells, and Müller glia, we generated microglial *Pparα* conditional knockout mice (PCKO, Figure 2A) and microglia-specific *Pparα* transgenic mice (PCTG, Figure 2B). Retinal thickness was quantified with OCT. Retinal thickness was decreased in PCKO mice relative to age-matched WT or PCTG mice (Figure 2C,D).

### 3.2. Increased Microglial Activation in Diabetic Retinas with Microglia-Specific Pparα Ablation

To measure microglia cell numbers in PCKO and PCTG mice, retinal microglia were stained with an anti-Iba1 antibody and counted in mouse eye sections. Interestingly, distinct from the effect in global *Pparα*^−/−^ mice, retinal microglial density did not differ between non-diabetic PCKO, PCTG, and WT animals at 8 months of age. However, in STZ-induced diabetic mice (8 months of age with diabetes duration 6 months), retinal microglial number was robustly increased in diabetic PCKO (PCKO-DM) retinas compared to WT diabetic (WT-DM) retinas (Figure 3A,B). 

Sub-retinal microglial accumulation is associated with neurodegeneration and retinal inflammation [35]. Sub-retinal microglia were labeled and quantified in RPE/Choroid flat mounts. Relative to WT-NDM animals, sub-retinal microglial cell density was increased in WT-DM mice. Microglial counts were further increased in PCKO-DM sub-retinal regions compared with WT-DM (Figure 3C). These findings demonstrated that loss of *Pparα* function contributes to subretinal microglial migration in diabetes.

### 3.3. Altered Retinal Function in Diabetic Microglia-Specific Pparα Conditional Knockout Mice 

To further evaluate retinal function in diabetic mice, electroretinogram (ERG) was used. The rod cell is the major photoreceptor type in mouse retinas [36]. Rod a-wave and b-wave amplitudes were decreased in WT-DM mice relative to WT-NDM mice (Figure 3A,B). Further, rod a-wave and b-wave amplitudes decreased in PCKO-DM mice relative to WT-DM mice (Figure 4A,B). Contrastingly, microglia-specific *Pparα* overexpression attenuated rod a-wave and b-wave amplitude reductions relative to WT-DM mice (Figure 4A,B). Photopic (Cone) ERG amplitude declined in diabetic PCKO mice, but not in WT-DM mice relative to non-diabetic mice (Figure 4C,D). These findings indicated that microglial *Pparα* deletion aggravated diabetes-induced neurodegeneration, and microglial *Pparα* overexpression alleviated diabetes-induced neurodegeneration.

### 3.4. Microglial Pparα Deletion Exacerbated Retinal Pericyte Loss in Diabetes

Pericyte loss is an early hallmark of DR that directly exacerbates vascular dysfunction [37,38]. To determine the regulatory role of microglial PPARα in the integrity of the retinal neurovascular unit, pericyte density was quantified (Figure 5). Diabetes significantly decreased retinal pericyte numbers in WT-DM mice relative to WT-NDM. Pericyte number was further decreased in PCKO-DM mice relative to WT-DM mice (Figure 5B). Contrastingly, diabetic pericyte loss was alleviated in PCTG-DM mice relative to WT-DM mice (Figure 5B). 

### 3.5. Pparα Knockdown Altered Human Microglial Metabolic Profile 

PPARα regulates multiple genes in glucose and lipid metabolism [39,40]. To investigate the association between PPARα metabolic regulation and microglial functional alteration in diabetes, the metabolic profile (Mitochondrial function and glycolytic rate) of HMC3 cells with and without *Pparα* knockdown (*Pparα* KD) was analyzed using a Seahorse extracellular flux analyzer. 4-Hydroxy-2-Nonenal (4-HNE) is a lipid peroxidation product that accumulates in the serum, plasma, body fluid, and tissues of diabetic patients and animal models. 4-HNE elevation is a biomarker for organ damage in diabetes and diabetic complications [41]. Previous studies have identified increased 4-HNE levels in the retina of diabetic patients and mouse models, which contributes to the pathogenesis of DR [42,43,44]. In this study, HMC3 cells were exposed to 4-HNE for 24 h following siRNA silencing with *Pparα* KD. The knockdown efficiency is presented in Appendix A. Under 4-HNE stress, *Pparα* KD impaired mitochondrial function and increased the basal glycolytic rate, as indicated by decreased basal oxygen consumption, maximal respiration, ATP production in Mito stress test analysis, and increased glycoPER value in Glycolytic rate analysis relative to cells transfected with scrambled siRNA (Figure 6A–E). These observations suggest that under diabetic stressors, *Pparα* ablation shifts microglial energy metabolism from oxidative phosphorylation to glycolysis.

Primary microglia were cultured from WT and *Pparα*^−/−^ mice to further test this finding. *Pparα* deficiency resulted in decreased mitochondrial oxidative phosphorylation (Figure 6F,G) and elevated glycolytic rate (Figure 6H) in *Pparα*^−/−^ microglia relative to WT microglia under 4-HNE stress. 

### 3.6. Pparα Deficiency Induced Microglial Pro-Inflammatory Polarization in Diabetic Conditions

To further determine if *Pparα* ablation was associated with microglial functional changes, expression levels of the pro-inflammatory regulator Stimulator of Interferon Genes (STING) and inflammatory cytokine Tumor necrosis factor-alpha (TNFα) were measured in primary microglial cells using Western blot analysis. Protein levels of STING and TNFα were increased in *Pparα*^−/−^ microglia relative to WT microglia (Figure 7A,B), indicating pro-inflammatory activation in *Pparα*^−/−^ microglia under normal conditions. Next, we determined if exposure to the diabetic stressor 4-HNE affected microglial *Pparα* expression. HMC3 cells were treated with 4-HNE, and PPARα protein level was measured by Western blot, revealing that 4-HNE treatment decreased PPARα protein levels in human microglia (Figure 7C).

Finally, to determine if the microglial STING pathway was activated in diabetic retinas, eyecup sections were co-stained for STING and microglial marker CD11b (Figure 7D). Microglial STING expression was quantified as the percentage STING positive area in each microglial cell. Retinal microglial STING expression was increased in PCKO-DM mice relative to WT-DM mice and PCTG-DM mice (Figure 7E). 

## 4. Discussion

Dysregulated metabolism and chronic inflammation contribute to the pathology of diabetic complications, including DR. Therefore, elucidating the relationships between metabolism, inflammation, and retinal pathologies is critical for developing therapeutic strategies to treat DR. In the present study, we investigated the role of PPARα, a central metabolic regulator, in retinal microglial activation and inflammation, and its effects on DR.

Microglial density and morphology are used as metrics of the retinal inflammatory status [45,46]. Increased microglial density and ameboid morphology are associated with microglial inflammatory responses [34]. Previous reports have identified a protective role for PPARα in neuroinflammation of the brain. In primary brain microglia, treatment with PPARα agonists negatively regulates LPS-stimulated pro-inflammatory cytokine expression [47,48]. In the radiation-induced brain injury model, administration of a PPARα agonist suppressed the inflammatory response by decreasing TNFα and IL1β levels [23]. For the first time, the present study revealed the role of PPARα in retinal microglial activation. Global *Pparα* knockout increased retinal microglial density, decreased microglial cell processes, and enlarged microglial soma size, which was suggestive of microglial activation. Furthermore, expression levels of STING and the pro-inflammatory cytokine TNFα were increased in *Pparα*^−/−^ microglia. The STING signaling pathway, which has traditionally been associated with the innate immune response to infection, has recently been identified as a central mediator of sterile inflammation in disease states [49,50]. Our findings demonstrate that PPARα plays a pivotal role in inhibiting microglial activation, partly by suppressing STING activation. 

Importantly, microglial density was increased in global *Pparα*^−/−^ mice without diabetes. However, this phenotype was not present in non-diabetic microglia-specific *Pparα* (PCKO) mice. We speculate that in global *Pparα*^−/−^ mice, pan-depletion of *Pparα* in all retinal cell types could cause complex alterations in the retinal microenvironment under normal conditions, including a glycolytic shift accompanied by impaired mitochondrial function in photoreceptors, central regulators of the retinal metabolic ecosystem [30,51]. On the other hand, in diabetes, microglia specific *Pparα* depletion exacerbated microglial activation.

Microglia activation and its contributions to neuronal dysfunction in brain disease are well-known in the contexts of Alzheimer’s disease, Parkinson’s disease, and multiple sclerosis [52,53]. Activated microglia release inflammatory factors and cytokines to promote neurodegeneration in the brain [54]. Diabetes is a chronic systemic disease, which induces neuronal dysfunction via an inflammatory cascade in the central nervous system [55,56,57,58,59]. In this study, retinal neurodegeneration by ERG recording in WT mice is consistent with prior studies in diabetic animals [60,61]. Conditional knockout of *Pparα* in the microglia aggravated, whereas conditional *Pparα* overexpression ameliorated diabetes-induced retinal dysfunction, suggesting the protective effect of microglial PPARα against diabetic neurodegeneration. Together, these observations indicate the therapeutic potential of targeting the microglial PPARα-associated cascade to treat neurodegeneration in DR. Indeed, we have previously reported that the PPARα agonist fenofibrate protected the retina against ERG decline and retinal degeneration in diabetic mice [25]. The present study provided further insight into prior findings, underscoring the specific role of PPARα in suppressing microgliosis-associated retinal dysfunction.

In addition to neuronal function damage, pericyte loss is a hallmark pathological feature of early DR, contributing significantly to retinal vascular pathologies, such as vascular leakage [38]. Our previous study demonstrated that PPARα alleviated retinal pericyte loss in diabetes [24]. Here, we observed that in diabetes, retinal pericytes were significantly decreased in both WT and PCKO mice. Interestingly, microglia-specific *Pparα* overexpression (PCTG) alleviated diabetes-induced pericyte loss. These observations indicate a specific protective role of microglial PPARα in maintaining integrity and function of the neurovascular unit. The relationship between microglial activation and pericyte apoptosis has been reported previously [62,63,64]. Gardiner et al. identified that in retinas from diabetic patients, juxtavascular microglia penetrate the basement membrane to scavenge and phagocytose pericytes [65]. This process is regulated by increased microglia-mediated inflammation in diabetic conditions. The findings of the present study emphasize the protective role of microglial PPARα in suppressing DR-induced pericyte loss. Vascular leakage, pericytes loss, and leukostasis are the significant characteristics of DR. In DR, activated microglia activation exacerbates angiogenesis-related phenotypes by secreting pro-inflammatory cytokines, which can induce retinal vascular pathologies such as vascular leakage [66]. Our results suggest that, not only pericyte loss but also vascular leakage could be aggravated in diabetic PCKO mice and alleviated in PCTG diabetic mice, compared with diabetic WT mice. 

Prior studies of PPARα agonists in microglial inflammation have focused on its role as a transcription factor of cytokines [22,67,68]. However, despite the role of PPARα as a central metabolic regulator, the relationship between PPARα-mediated metabolic changes and microgliosis has not previously been evaluated. Here, we demonstrated that knocking down PPARα expression in microglia induced metabolic reprogramming under diabetic stress conditions. Specifically, *Pparα* ablation decreased mitochondrial respiration while concomitantly increasing glycolysis, a metabolic change characteristic of microglial activation [12,14]. Consistently, we identified that under the same conditions, the STING pathway was activated, and pro-inflammatory cytokine expression was increased. 

Microglial function is regulated in part by metabolic reprogramming, in which a shift towards glycolytic metabolism increases inflammatory processes. Blocking microglial glycolysis with 2-deoxyglucose, a glycolytic inhibitor, reverses the LPS-induced inflammatory response [14,69], while suppressing mitochondrial respiration with mitochondrial toxins induces microgliosis [70,71]. Further, protection of mitochondrial function alleviates microglia-mediated neuroinflammation and increases microglial Aβ clearance in Alzheimer's disease [72]. These observations indicate that protection of mitochondrial function is associated with microglial quiescence or suppression of inflammation, while a glycolytic shift could trigger pro-inflammatory microglial polarization. Therefore, we speculate that metabolic aberrations caused by loss of PPARα function in DR [26] contribute to inflammatory microglial polarization, initiating neuronal dysfunction and damage to the neurovascular unit. 

## 5. Conclusions

The present study revealed the role of PPARα in regulating retinal microglia metabolism and activation in the context of diabetes. PPARα has protective effects against microglia-induced neuronal dysfunction and neurovascular unit damage in the diabetic retina. 

## Figures and Tables

**Figure 1 cells-11-03869-f001:**
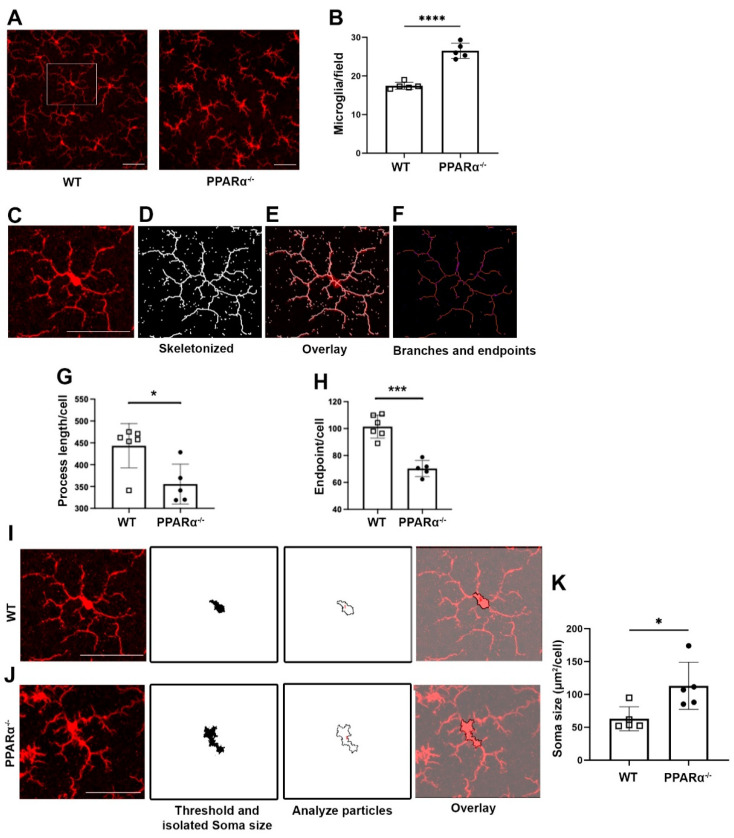
Microglial activation in *Ppara*^−/−^ retinas. (**A**) Representative flat-mounted retina from 9-month-old Wild-type (WT) and *Pparα*^−/−^ mice stained with anti-Iba1 antibody. (**B**) Microglia counts in WT and *Pparα*^−/−^ retinas. (**C**–**F**) Microglial morphological analysis with Fiji ImageJ. Images were skeletonized (**D**) and overlaid with original images (**E**) to verify accuracy. Branches and intersections were labeled (**F**). (**G**,**H**) Quantification of microglial branch length (**G**) and endpoints (**H**) in WT and *Ppara*^−/−^ retinas. (**I**,**J**) Cell bodies were isolated from images of WT (**I**) and *Pparα*^−/−^ (**J**) retinas. (**K**) Quantification of WT and *Pparα*^−/−^ microglial soma size. *n* = 5–6. Data are presented as means ± SD. * *p* < 0.05, *** *p* < 0.001, **** *p* < 0.0001, Student’s *t*-test. Scale bar: 50 μm.

**Figure 2 cells-11-03869-f002:**
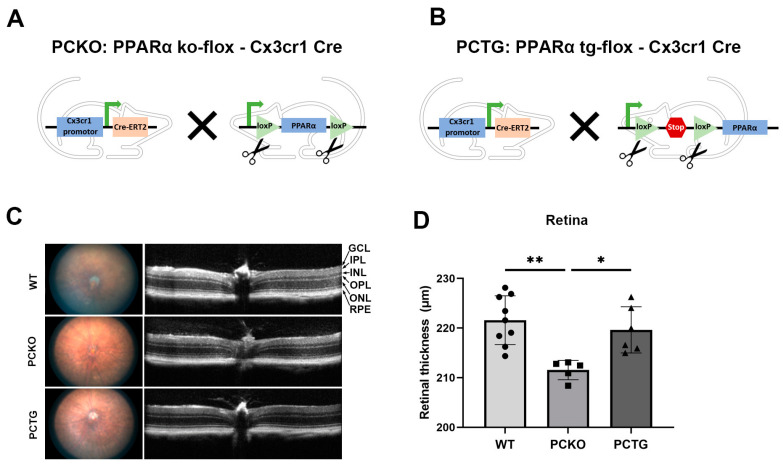
Decreased retinal thickness in PCKO mice. (**A**,**B**) Diagrams of PCKO-Cx3Cr1Cre (**A**) and PCTG-Cx3Cr1Cre (**B**) mouse lines constructs. (**C**) Representative retina fundus and OCT images of non-diabetic WT, PCKO, and PCTG mice (8 months of age). (**D**) Total retinal thickness quantified by OCT. *n* = 5–9. * *p* < 0.05, ** *p* < 0.01, One-way ANOVA with Tukey’s posthoc comparison. Abbreviations: IPL: inner plexiform layer, INL: inner nuclear layer; OPL: outer plexiform layer, ONL: outer nuclear layer; GCL: ganglion cell layer; RPE: retinal pigment epithelium, WT: wild-type, PCKO: Microglial *Pparα* conditional knockout, PCTG: Microglial *Pparα* conditional transgenic mice.

**Figure 3 cells-11-03869-f003:**
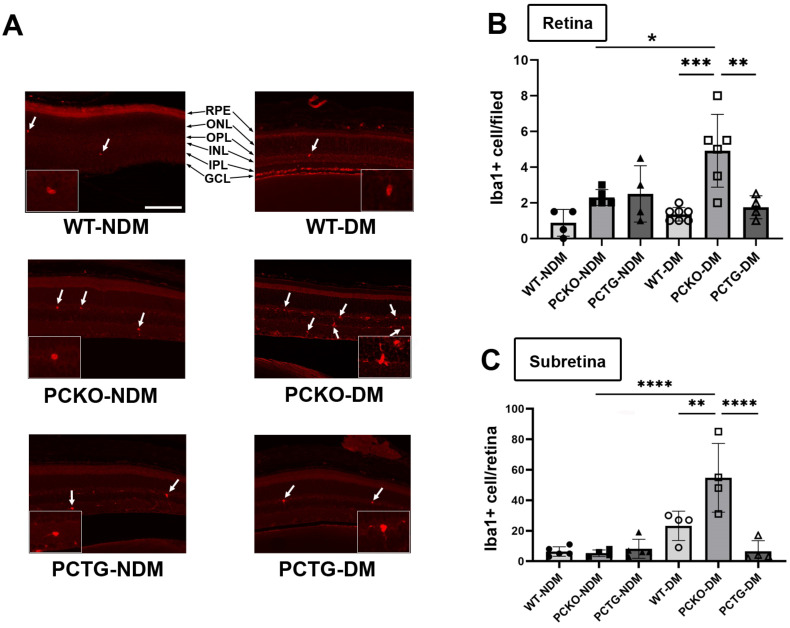
Increased microglial density in diabetic PCKO mice. (**A**) Representative immunostaining images with Iba1^+^ microglia (arrows) in retina sections. Scale bar: 100 μm. (**B**) Quantification of retinal microglia density. *n* = 4–7 mice. (**C**) Subretinal microglial density on RPE/Choroid flat mounts, *n* = 4–7 mice. One-way ANOVA with Tukey’s posthoc comparison. Data are presented as mean ± SD. * *p* < 0.05, ** *p* < 0.01, *** *p* < 0.001, **** *p* < 0.0001. Abbreviations: IPL: inner plexiform layer, INL: inner nuclear layer; OPL: outer plexiform layer, ONL: outer nuclear layer; GCL: ganglion cell layer; RPE: retinal pigment epithelium, DM: diabetes mellitus, NDM: non-diabetes, WT: wild-type, PCKO: Microglial *Pparα* conditional knockout, PCTG: Microglial *Pparα* conditional transgenic mice.

**Figure 4 cells-11-03869-f004:**
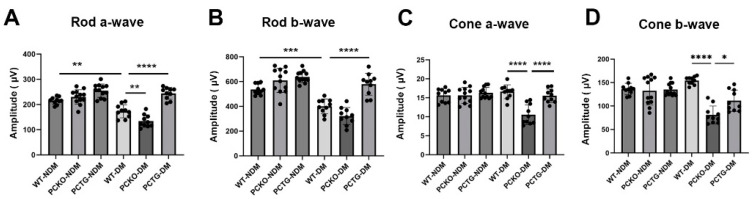
Impaired retinal function in diabetic PCKO mice. A-wave (**A**,**C**) and b-wave (**B**,**D**) amplitudes of scotopic ERG (Rod, (**A**,**B**)) and photopic ERG (Cone, (**C**,**D**)) were recorded in diabetic mice and age-matched non-diabetic controls at 6 months after diabetes onset. *n* = 10–12. Data are presented as mean ± SD. * *p* < 0.05, ** *p* < 0.01, *** *p* < 0.001, **** *p* < 0.0001, one-way ANOVA with Tukey’s posthoc comparison. Abbreviations, DM: diabetes mellitus, NDM: non-diabetes, WT: wild-type, PCKO: Microglial *Pparα* conditional knockout, PCTG: Microglial *Pparα* conditional transgenic mice.

**Figure 5 cells-11-03869-f005:**
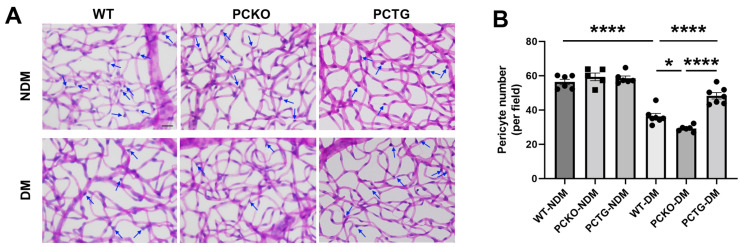
Exacerbated pericyte loss in diabetic PCKO mice. (**A**) Retinal vasculature was prepared by trypsin digestion followed by periodic-acid-Schiff staining. Scale bar: 20 μm. Pericytes (arrows) were counted and compared. Scale bar: 20 μm. (**B**) Quantitative analysis of retinal pericyte density. *n* = 5–6. Data are presented as mean ± SD. * *p* < 0.05, **** *p* < 0.0001, one-way ANOVA with Tukey’s posthoc comparison. Abbreviations, DM: diabetes mellitus, NDM: non-diabetes, WT: wild-type, PCKO: Microglial *Pparα* conditional knockout, PCTG: Microglial *Pparα* conditional transgenic mice.

**Figure 6 cells-11-03869-f006:**
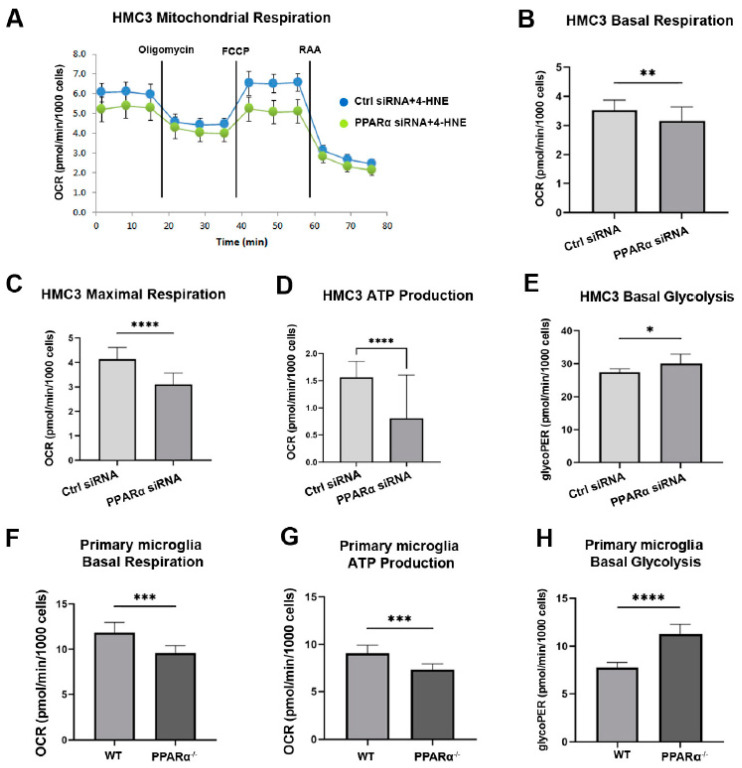
Altered metabolic profile with *Pparα* deletion. (**A**) Representative real-time trace of oxygen consumption rates (OCR) using a Mito stress test in human microglia exposed to 4-HNE for 24 h following *Pparα* siRNA knockdown. (**B**–**D**) Quantification of basal OCR (**B**), Maximal respiration (**C**), and ATP production (**D**) in *Pparα* siRNA knockdown HMC3 cells relative to HMC3 cells transfected with scramble siRNA, *n* = 5–8. Student’s *t*-test. * *p* < 0.05, ** *p* < 0.01. (**E**) Glycolytic proton efflux rate (glycoPER) measurement obtained from microglia cells exposed to 4-HNE for 24 h following *Pparα* knockdown and subject to the glycolytic rate assay. *n* = 5–8. * *p* < 0.05, Student’s *t*-test. (**F**,**G**) Mito stress test assay in mouse WT and *Pparα*^−/−^ mouse primary microglial cells exposed to 4-HNE for 24 h. Quantification of basal OCR (**F**) and ATP production (**G**). (**H**) Glycolytic rate assay in mouse WT and *Pparα*^−/−^ mouse primary microglial cells exposed to 4-HNE for 24 h. *n* = 5–8. Student’s *t*-test. Data are presented as mean ± SD. *** *p* < 0.001, **** *p* < 0.0001.

**Figure 7 cells-11-03869-f007:**
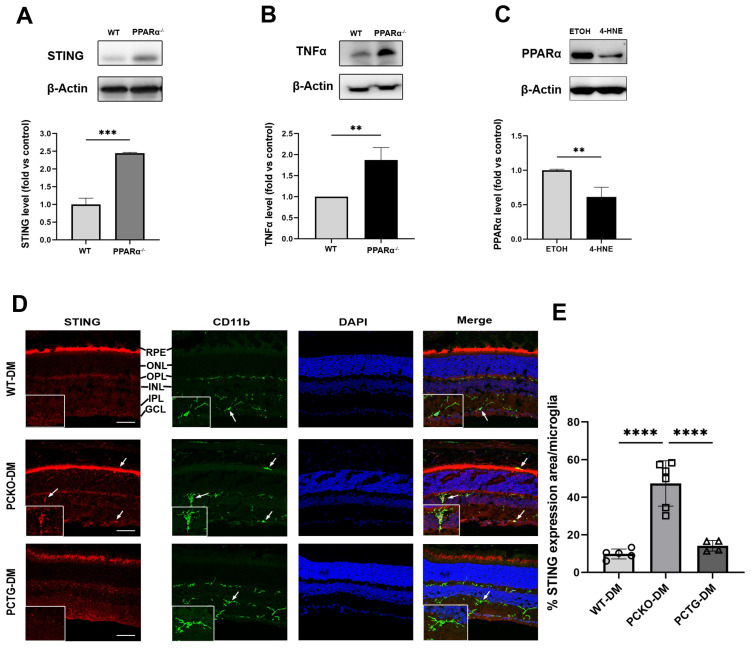
Increased cytokine release in microglia with *PPARα* deletion. (**A**,**B**) STING and TNFα protein levels in primary WT and *Pparα*^−/−^ microglial cells. *n* = 3. ** *p* < 0.01, *** *p* < 0.001, Student’s *t*-test. (**C**) PPARα protein levels as quantified by Western blot analysis in HMC3 cells following 24 h 4-HNE exposure. *n* = 3. ** *p* < 0.01, Student’s *t*-test. (**D**) STING/CD11b co-immunostaining (arrows) in WT-DM, PCKO-DM, and PCTG-DM retinas. Scale bar: 50 µm. (**E**) Quantification of STING expression per microglial cell. *n* = 4−6. **** *p* < 0.0001, One-way ANOVA with Tukey’s posthoc comparison. Data are presented as mean ± SD.

## Data Availability

All data are included in this manuscript.

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
