# Peer review of "The Protective Role of Microglial PPARα in Diabetic Retinal Neurodegeneration and Neurovascular Dysfunction"

_cells, 2022, doi:10.3390/cells11233869_

Round 1
Reviewer 1 Report
In this paper, the authors investigated the protective role of microglial PPARα in diabetic retinal neurodegeneration and neurovascular dysfunction. Under diabetic conditions, PCKO mice (PPARα conditional knockout mice with increased microglial density) had reduced ERG responses and increased pericyte loss, while PCTG mice (PPARα transgenic mice with reduced microglial density) showed alleviated diabetes-induced retinal dysfunction. In cultured microglial cells, PPARα loss of function caused a metabolic switch from oxidative phosphorylation to glycolysis in a 4-HNE-induced diabetic condition. PPARα deficiency also increased microglial STING and TNF-α expression.
The paper covers an interesting topic, but needs some revision on the following points:
1. Abstract
- Line 16-17: I would suggest moving this sentence down in the abstract. I think it is more logic to first describe the results in the non-diabetic knockout and transgenic mice, followed by the results in the diabetic mice.
2. Material and methods
- To make it more clear, I would suggest to first describe all the read-outs performed in the mice, followed by all the in vitro work.
- Information on sacrificing the animals and timepoints of the in vivo read-outs is missing in the M&M section.
- Please add some more information on how the analysis of the ERG amplitudes was performed.
3. Results
- Can the authors explain why different ages of animals were used for the various read-outs? This makes it difficult to make a connection between the different findings.
- Line 215-217: Retinal thickness measurements – this is a confusing paragraph:
o Please specify if these retinal thickness measurements were performed in diabetic or non-diabetic mice. According to the title of paragraph 3.2 it refers to diabetic mice, but according to the figure to non-diabetic.
o If this was done in non-diabetic mice, I would suggest moving this part to section 3.1 and if performed in diabetic mice, I would suggest to move it to section 3.3 or 3.4, where other diabetic characteristics are investigated.
o If only performed in non-diabetic mice, it would be very interesting to see the same measurements in diabetic mice as well.
- Line 218-232: Microglia cell density in diabetic mice
o In this paragraph, the authors do describe Iba1 staining on flatmounts, but no results are shown from this. Please change or add the data.
o What can be the explanation why no significant changes in retinal microglia density were seen between WT and PCKO mice (since this was present in the global ko mice)? Can this be due to different age of mice? Can you also provide more details for the possible explanation that you highlight in the discussion, e.g. What do you mean with “complex retinal environmental stimulators”?
o How do you explain that no difference in microglia density in the retina is observed between WT-NDM and WT-DM, since diabetes on its own in WT mice should already lead to an increased number of inflammatory cells. In the subretina there seems to be an increase, but is this significantly different?
o Was the microglial morphology (cfr. Section 3.1) also investigated in the diabetic mice (if analysis was performed on flatmounts)? This would be very interesting to see if the differences are also present under diabetic conditions.
o It would be good to also indicate the different retinal layers on the images of the Iba-1 stained retinal sections.
4. Discussion
- In the introduction macrophage polarization is introduced, is there a link described between this and PPARα. If so, please add this information in the discussion.
- Line 327-329: “We demonstrated that PPARα global knockout resulted in an increased retinal microglial density, decreased cell processes, and enlarged soma size in the diabetic condition.” This statement is not correct, since this was not demonstrated in diabetic conditions, please add data for this or change the sentence.
- Besides thickening, inflammation & pericyte loss, leakage is also a very important disease hallmark of DR/DME. It would be very interesting to see how the leakage would behave in diabetic PCKO and PCTG mice. Or at least it would be good if the authors can speculate on this in the discussion.
- Line 391-394: You can move that paragraph to the "5. Conclusion section"
5. Minor changes
- Information on the eye drops used for inducing pupil dilation is missing in M&M section.
- Details on how the flatmounts and sections were mounted, is missing in M&M section.
Author Response
Dear Professor,
We sincerely appreciate the your professional comments and suggestion.
Please check the attachment file for our point-by-point responses.

Reviewer 2 Report
The authors present an interesting study on the role of PPARa in microglia cells during diabetic retinopathy. The authors show experiments utilizing microglia specific PPARa ko and overexpressing (TG). The data seems convincing and would add to our understanding of DR pathogenesis.
There are two major considerations from this Reviewer to the authors:
1) It would be nice to see molecular data on the PCKO and PCTG mice that would explain the differences in their phenotypes. Much of the mechanistic work is performed on microglia cell cultures and isolated microglia from global PPARa ko mice.
2) How does the PCKO and PCTG mice resemble the global PPARa ko mice after they were made diabetic? Are they similar or less severe? That would be nice to know since it appears PPARa is important in microglia and could show a significant role of these cells in the STZ-induced diabetic retinopathy model.
Some other specific considerations are as follows:
Results:
3.1
How many microglia were assessed per mouse in this study?
Was the localization of microglia different in the PPARa ko mice?
Forgot to add ****, P<0.0001 to the figure legend of Figure 1.
3.2
The authors noted that there was retinal thinning in the PCKO mice at 8 months of age. Was there a retinal thinning in the Ppara-/- mice included in Figure 1?
Were all mice treated with tamoxifen or only the PCKO and PCTG mice treated with tamoxifen?
In panel G, is WT-DM statistically different from PCTG-DM mice? If not, then the authors should remove statement (lines 230-231) as the presence of PPARa prevents diabetes-induced microglia migration.
3.3
This reviewer would recommend making a figure that is devoted to the ERG results from this study (separate Panels A-D from E-F).
Also, discussion of the ERG results could be better. There is a lot of data being presented and it is not well discussed.
3.4
Does this occur in the PPARa knockout mice?
Make panels E-F a separate figure.
3.5
It is not clear why 4-HNE is being used in these studies nor is it described in the materials and methods of the paper.
What was the efficiency of the siRNA knockdown of PPARa in the cell culture experiments?
Were there differences between control and PPARa knockdown cells not treated with 4HNE from the seahorse studies?
Why weren’t microglia isolated from PCTG and PCKO mice for these studies?
I would also suggest making a figure of only panels A-H and a separate figure with I-J.
3.6
The authors should examine if there is higher STING and TNFalpha in the retinas of the PCKO (and vice versa in the PCTG) mice after DM treatment to confirm their in vivo findings….
Discussion
Lines 327-329: PPARa global knockouts used in this study were mentioned to be diabetic but in the results section they were not, were these mice diabetic or not? This sentence or the results should be revised accordingly.
Lines 335-338: Specify what cell types these could be in the retina
Line 338: Remove the first In diabetes in this sentence
Author Response
Dear Professor
We sincerely appreciate your professional comments.
We have uploaded our point-by-point responses in the attached file.

Round 2
Reviewer 1 Report
All my comments and suggestions are addressed.